# Suppression of Angiogenesis by Targeting Cyclin-Dependent Kinase 7 in Human Umbilical Vein Endothelial Cells and Renal Cell Carcinoma: An In Vitro and In Vivo Study

**DOI:** 10.3390/cells8111469

**Published:** 2019-11-19

**Authors:** Chung-Sheng Shi, Kuan-Lin Kuo, Mei-Sin Chen, Po-Ming Chow, Shing-Hwa Liu, Yu-Wei Chang, Wei-Chou Lin, Shih-Ming Liao, Chen-Hsun Hsu, Fu-Shun Hsu, Hong-Chiang Chang, Kuo-How Huang

**Affiliations:** 1Graduate Institute of Clinical Medical Sciences, College of Medicine, Chang Gung University, Taoyuan 333, Taiwan; csshi@mail.cgu.edu.tw (C.-S.S.); leo7771879@yahoo.com.tw (M.-S.C.); 2Division of Colon and Rectal Surgery, Department of Surgery, Chiayi Chang Gung Memorial Hospital, Chiayi County 613, Taiwan; 3Department of Urology, College of Medicine, National Taiwan University, and National Taiwan University Hospital, Taipei 100, Taiwan; antibody0123@gmail.com (K.-L.K.); meow1812@gmail.com (P.-M.C.) andy79122@hotmail.com (Y.-W.C.); sanguine444@gmail.com (S.-M.L.); chanhsun.hsu@gmail.com (C.-H.H.); changhong@ntu.edu.tw (H.-C.C.); 4Graduate Institute of Toxicology, College of Medicine, National Taiwan University, Taipei 100, Taiwan; shinghwaliu@ntu.edu.tw; 5Department of Pathology, College of Medicine, National Taiwan University, and National Taiwan University Hospital, Taipei 100, Taiwan; weichou8@ms52.hinet.net; 6Graduate Institute of Clinical Medicine, College of Medicine, National Taiwan University, Taipei 106, Taiwan; fs_hsu@outlook.com; 7Department of Urology, Taipei City Hospital Heping Fuyou Branch, Taipei 100, Taiwan

**Keywords:** THZ1, cyclin-dependent kinase 7, angiogenesis, endothelial cell, renal cell carcinoma

## Abstract

Cancer cells rely on aberrant transcription for growth and survival. Cyclin-dependent kinases (CDKs) play critical roles in regulating gene transcription by modulating the activity of RNA polymerase II (RNAPII). THZ1, a selective covalent inhibitor of CDK7, has antitumor effects in several human cancers. In this study, we investigated the role and therapeutic potential of CDK7 in regulating the angiogenic activity of endothelial cells and human renal cell carcinoma (RCC). Our results revealed that vascular endothelial growth factor (VEGF), a critical activator of angiogenesis, upregulated the expression of CDK7 and RNAPII, and the phosphorylation of RNAPII at serine 5 and 7 in human umbilical vein endothelial cells (HUVECs), indicating the transcriptional activity of CDK7 may be involved in VEGF-activated angiogenic activity of endothelium. Furthermore, through suppressing CDK7 activity, THZ1 suppressed VEGF-activated proliferation and migration, as well as enhanced apoptosis of HUVECs. Moreover, THZ1 inhibited VEGF-activated capillary tube formation and CDK7 knockdown consistently diminished tube formation in HUVECs. Additionally, THZ1 reduced VEGF expression in human RCC cells (786-O and Caki-2), and THZ1 treatment inhibited tumor growth, vascularity, and angiogenic marker (CD31) expression in RCC xenografts. Our results demonstrated that CDK7-mediated transcription was involved in the angiogenic activity of endothelium and human RCC. THZ1 suppressed VEGF-mediated VEGFR2 downstream activation of angiogenesis, providing a new perspective for antitumor therapy in RCC patients.

## 1. Introduction

Dysregulated angiogenesis plays a crucial role in tumor growth and metastasis [1]. Vascular endothelial growth factor (VEGF)-mediated VEGF receptor 1 and 2 (VEGFR1/2) signaling plays critical roles in the angiogenic activity of the endothelium [2]. Several antiangiogenic agents that neutralize or block VEGFR have been approved for treating some cancer entities, including metastatic renal cell carcinoma (RCC), and have become the mainstay of metastatic RCC therapy [3].

Eukaryotic RNA polymerase II (RNAPII)-catalyzed gene transcription is critical and is facilitated by multiple transcription factors [4,5]. A deregulated transcriptional machinery has been found in most malignancies [6], and drugs directly targeting the transcription machinery can inhibit tumor growth [7]. At the stage of transcription initiation, cyclin-dependent kinase (CDK) 7 phosphorylates the RNAPII C-terminal domain (CTD) at serine 5 and 7 to participate in the initiation and elongation of transcription [8]. CDKs are important in regulating cell cycle progression and gene transcription, and CDK deregulation has been observed in human malignancies. At least 20 CDKs in mammalian cells have been identified. Briefly, the functions of CDK7–13 are linked to transcription, and CDK1, 2, 4, and 6 are associated with cell cycle regulation. Among them, CDK7 plays a dual role of regulating both cell cycle and transcription. Cytosolic CDK7 forms a heterotrimeric complex and functions as a CDK1/2-activating kinase [9,10,11]. Nuclear CDK7 associates with the kinase core of the RNAPII transcription factor complex by phosphorylating the CTD of RNAPII, after which gene transcription can be initiated [12,13]. CTD phosphorylation of CDK7 is critical for modulating cell proliferation, cell cycle progression, and cell apoptosis. Several small molecular inhibitors of CDKs are currently being developed as antitumor therapies [14]. A previous study indicated that first-generation nonselective CDK inhibitors, such as flavopiridol, are potent tumor inhibitors that directly inhibit CDK1, 2, 4, and 7 by suppressing phosphorylation, and subsequently, activation of CDKs and cyclins [15]. However, these drugs have also presented substantial toxicity and side effects in early clinical trials. Developing more selective inhibitors of CDKs is a promising approach for cancer treatment. Studies investigating the effect of CDK regulation on angiogenesis are limited [16,17,18]. A recent study reported that roscovitine, a CDK inhibitor, suppresses the angiogenic activity of endothelial cells in vitro and in vivo by targeting CDK2, 5, 7, and 9 [19]. Furthermore, CDK5 has been demonstrated to be an angiogenic regulator that can control the migration and angiogenic activity of endothelial cells in vitro and in vivo [20]. 

Approximately 50% patients with clear cell RCC carry *Von Hippel–Lindau* (*VHL*) gene mutation and deregulated hypoxia-inducible factor-related downstream genes, including *VEGF* [21]. Therapies targeting VEGF pathway inhibitors have been approved for treating advanced or metastatic cancer. THZ1, a selective covalent inhibitor of CDK7, targets the cysteine residue located outside the canonical kinase domain and covalently inhibits CDK7 [22,23], thereby leading to the effective inhibition of the growth of several tumors [22,24,25]. However, the effect of THZ1 on angiogenesis and RCC remains unclear. The antitumor effects of THZ1 have been reported in neuroblastoma, small cell lung cancer, and triple-negative breast cancer [22,26,27]. 

In this study, we evaluated the role of CDK7 in regulating the angiogenic activity of human umbilical vascular endothelial cells (HUVECs), as well as the antiangiogenic and antitumor effects of THZ1 on RCC cells.

## 2. Materials and Methods

### 2.1. Reagents and Antibodies

THZ1 (#M5228) was purchased from AbMole BioScience, Inc. (Houston, TX, USA). Antibodies against various proteins for Western blot analyses, such as CDK7, RNAPII, RNAPII pS5, RNAPII pS7, cleaved poly ADP ribose polymerase (PARP), cleaved caspase-3, cleaved caspase-7, VEGFR2, CD31, and VEGF, were obtained from Cell Signaling Technology (Danvers, MA, USA). The β-actin antibody was purchased from GeneTex (Irvine, CA, USA), and the α–tubulin antibody was purchased from Santa Cruz Biotechnology (Santa Cruz, CA, USA). All the other chemicals and reagents were obtained from Sigma-Aldrich (St. Louis, MO, USA), Merck Millipore (Billerica, MA, USA), and Invitrogen (Carlsbad, CA, USA). 

### 2.2. Cell Culture and siRNA Transfection

HUVECs and human RCC cell lines (786-O and Caki-2) were obtained from the Bioresource Collection and Research Center, Taiwan. The 786-O and Caki-2 cell lines were cultured in high-glucose Dulbecco’s modified eagle medium supplemented with 10% fetal bovine serum (FBS), penicillin (100 U/mL), and streptomycin (100 μg/mL). The HUVECs were cultured in complete M199 medium containing 20% FBS, endothelial cell growth supplement (Millipore, Billerica, MA, USA), penicillin (100 U/mL), and streptomycin (100 μg/mL) in the 0.1% gelatin (Sigma-Aldrich)-coated plate. The three types of cells were maintained at 37 °C in humidified air containing 5% CO_2_. All the other culture media and supplements were obtained from Invitrogen. Furthermore, in siRNA interfering experiment, HUVECs were cultured to 80% confluence in the gelatin-coated 6 cm diameter dishes in in complete M199 medium. After culture, cells were rinsed with serum-free M199 and transfected with siRNA (GenePharma, Shanghai, China) for nontargeting scramble (5′- UUGUACUACACAAAAGUACUG-3′) or CDK7 (5′-CUGAUCUAGAGGUUAUAAUTT-3′ and 5′- AUUAUAACCUCUAGAUCAGTT-3′; cdk-466) using Lipofectamine RNAiMAX (Invitrogen) according to the manufacturer’s instructions. After 24 h, the transfected HUVECs were subjected to Western blotting analysis for verifying CDK7 expression or harvested for tube formation assay for examining the effect of CDK7 in angiogenic activity of HUVECs.

### 2.3. Cell Proliferation Assay

Cell proliferation was determined through the water-soluble tetrazolium 1 (WST-1, 4-[3-(4-iodophenyl)-2-(4-nitrophenyl)-2H-5-tetrazolio]-1, 3-benzene disulfonate) assay (BioTools, Taipei, Taiwan). HUVECs were seeded into a gelatin-coated 96-well plate in complete M199 medium containing endothelial cell growth supplement and 20% FBS. After 18 h, the cells were incubated with or without VEGF (50 ng/mL; Invitrogen) and various concentrations of THZ1 (50, 100, 250, and 500 nM) in complete M199 for 24 or 48 h. After the indicated incubation periods, WST-1 (Roche Diagnostics, Vienna, Austria) was added to the cells according to the manufacturer’s protocol to measure the amount of formazan dye formed by metabolically active cells, which directly correlates to the number of viable cells in the culture. The data was expressed as proliferation (% of mock control).

### 2.4. Western Blotting 

After various treatments, cells from each cell line were washed with ice-cold phosphate-buffered saline (PBS) and lysed with cell lysis buffer (Cell Signaling Technology) on ice for 15 min followed by centrifugation at 14,000 rpm for 15 min at 4 °C. The clear supernatants were harvested, and protein concentrations were determined through the bicinchoninic acid protein assay (Thermo Fisher Scientific, Waltham, MA, USA). Equal quantities of each sample were resolved by sodium dodecyl sulfate-polyacrylamide gel electrophoresis and were then transferred to polyvinylidene fluoride membranes (Millipore). The membranes were blocked with 5% bovine serum albumin in Tris-buffered saline containing Tween 20 (TBST) for at least 1 h followed by overnight incubation with respective primary antibodies at 4 °C. The membranes were then washed three times with TBST for 10 min each and were incubated at room temperature for 1 h with horseradish peroxidase-conjugated secondary antibodies (GeneTex). After washing twice with TBST, antibody-bound membranes were visualized with enhanced chemiluminescence Western blot detection reagents (Millipore and Biotools). Further, to quantify protein levels in Western blot, the band intensities were analyzed by the Image J software (version 1.52q, NIH, Bethesda, MD, USA), and the fold change of protein level relative to their control was calculated and expressed as the numbers under the plots [28]. 

### 2.5. Transwell Migration Assay

HUVECs (2 × 10^4^ cells) in 100 μL M199 medium containing 5% FBS were loaded into the upper chambers of the Transwell (Corning Life Science, Corning, NY, USA) apparatus. VEGF (10 ng/mL), a potent chemotactic agent of endothelial cells, with or without various concentrations of THZ1 (50, 100, 250, and 500 nM) in 5% FBS-containing M199 were loaded in the lower chambers for HUVEC migration. After 4 h, the cells on the top surface of the membrane were scraped with a cotton swab. The migrated cells on the lower surface of the membrane were fixed with methanol for 30 min. Next, PBS washing was performed three times followed by crystal violet (1%) staining. We acquired photographs using microscope at 100× magnification for quantifying chemotaxis of HUVECs by counting three random fields in each well using Image J software. The migrated cells in each group were expressed as migration (% of mock control). 

### 2.6. In Vivo Matrigel Plug Assay 

The Matrigel plug assay was performed as described previously [29]. To further evaluate the angiogenic effect of THZ1 in vivo, liquid Matrigel (Corning Life Science) containing heparin (20 U) and VEGF (100 ng) with or without THZ1 (250 nM) were injected into FVB (Friend leukemia virus B) mice (*n* = 3, obtained from the Taiwan National Laboratory Animal Center). Matrigel with PBS plus heparin served as negative controls, whereas VEGF plus heparin was used as positive controls. Five days after administering the injection, the mice were sacrificed by CO_2_ asphyxiation. The Matrigel plugs were excised and photographed. For further analysis of angiogenesis, hemoglobin content in Matrigel plug was measured with hemoglobin colorimetric assay kit (Cayman Chemicals, Ann Arbor, MI, USA). Briefly, Matrigel plugs were weighted and then homogenized in 1 mL deionized H_2_O for 5–10 min on ice. After centrifugation, supernatants were collected and mixed with hemoglobin detector at room temperature for 15 min. After incubation, the concentration of hemoglobin in Matrigel plugs was calculated with a standard curve by measuring the absorbance at 560 nm according to manufacturer’s protocol. Furthermore, the calculated values were normalized with plug weight and expressed as g/dl hemoglobin per milligram Matrigel plug. 

### 2.7. Capillary Tube Formation Assay

Matrigel was thawed on ice at 4 °C, and then 10 µL was added to the wells of a µ-Slide angiogenesis plate (Ibidi GmbH, Gräfelfing, Germany) and polymerized for 30 min at 37 °C. HUVECs were harvested and resuspended in 10% FBS-containing M199 medium. The cell suspensions (40 µL each) containing VEGF (10 ng/mL) with or without THZ1 (250 nM) were plated onto Matrigel-coated wells, and the plates were maintained at 37 °C for 16–24 h. Each well was then photographed for further quantification of angiogenic activity of endothelial cells by measuring the formation of tubes on Matrigel using the Image J software.

### 2.8. In Vivo Xenograft Experiments 

Briefly, the cultured RCC cells were trypsinized, harvested freshly to ensure optimal cell viability, and resuspended in PBS. A total of 1 × 10^5^ 786-O and Caki-2 cells suspended in 200 μL serum-free medium and mixed with an equal volume of Matrigel were injected into the flank of NOD SCID (NOD.CB17-*Prkdc*^scid^/NcrCrl) mice (obtained from the Taiwan National Laboratory Animal Center). After tumor growth reached approximately 150 mm^3^, the mice were randomly assigned to the THZ1-treated and control groups. The THZ1-treated group (*n* = 8) was intraperitoneally injected with THZ1 (10 mg/kg) in dimethyl sulfoxide (DMSO) every 2 days for 28 days, whereas the control group (*n* = 7) was administered with DMSO. Tumor volumes were measured with calipers twice per week. Tumor volume was calculated using the formula V = LD × (SD)^2^/2, where V is the tumor volume, LD is the longest tumor diameter, and SD is the shortest tumor diameter [28]. All studies involving animal experiments, animal care, and experimental procedures were approved by the National Taiwan University College of Medicine and College of Public Health Institutional Animal Care and Use Committee (IACUC, No. 20160280 and 20170268). All studies involving animals complied with the ARRIVE (Animal Research: Reporting of *In Vivo* Experiments) guidelines for the reporting of experiments involving animals. After 4 weeks, the mice were sacrificed by CO_2_ asphyxiation, and the tumors were surgically removed and photographed. For immunohistological analysis of tumors, the rat anti-mouse CD31 (platelet endothelial cell adhesion molecule-1) antibody (BD Biosciences, San Jose, CA, USA) was used for representing CD31 positive neo blood vessels inside tumors. The same specimens were co-stained with hematoxylin and eosin. Microvessel density (MVD) was calculated in tumor section by counting three highly vascularized fields in each section using the Image J software and expressed as MVD (% of DMSO control).

### 2.9. Statistical Analysis

GraphPad Prism^®^ 5 software was used to perform all data analysis (GraphPad, Inc., La Jolla, CA, USA). All data were expressed as mean ± standard deviation. The unpaired Student’s *t*-test was used to compare means of two independent samples. One-way ANOVA was used to compare means of two or more samples followed by the Bonferroni post hoc test—a *p* value less than 0.05 was considered as statistically significant.

## 3. Results

### 3.1. VEGF Enhances CDK7 and RNAPII Activation and Downstream RNAPII Phosphorylation at Serine 5 and 7 in HUVECs

VEGF, the primary regulator of angiogenesis, mediates the migration, proliferation, and capillary tube formation of endothelial cells through VEGFR1/2. The effect of VEGF on transcription initiation in HUVECs and CDK7-mediated RNAPII CTD phosphorylation at serine 5 and 7 remains unclear [8]. Our data revealed that VEGF stimulation enhanced CDK7 and RNAPII expressions and upregulated downstream RNAPII phosphorylation at serine 5 and 7 in HUVECs in a time-dependent manner (Figure 1). The results indicated that the CDK7-associated transcription machinery was involved in VEGF-activated angiogenesis of HUVECs.

### 3.2. THZ1 Dose-Dependently Suppresses the Proliferation and Chemotactic Migration of HUVECs 

CDK7-mediated transcription was implicated in the VEGF-activated endothelium. Therefore, the functional effect of CDK7 inhibition on the angiogenic activities of the endothelial cells was investigated using THZ1, a selective covalent CDK7 inhibitor. Figure 2A illustrates that VEGF stimulation significantly enhanced HUVEC proliferation after 24 and 48 h. THZ1 dose-dependently suppressed VEGF-activated endothelial cell proliferation after stimulation for 24 and 48 h. Furthermore, Figure 2B,C illustrates that VEGF significantly promoted the chemotactic mobility of HUVECs. THZ1 significantly inhibited the VEGF-activated mobility of the endothelial cells in a dose-dependent manner. The results revealed that CDK7 inhibition can interfere with VEGF-activated endothelial cell proliferation and mobility, indicating that CDK7-associated transcription was critically involved in VEGF-activated angiogenesis. 

### 3.3. THZ1 Induces Apoptosis and Suppresses VEGFR2 Expression and RNAPII CTD Phosphorylation at Serine 5 and 7 in HUVECs

We analyzed the apoptotic and angiogenic effects of CDK7 inhibition by treating HUVECs with various concentrations of THZ1 (100–500 nM) for 48 h. Figure 3 shows that THZ1 did not significantly affect the expression of CDK7, but significantly diminished RNAPII CTD phosphorylation at serine 5 and 7 in HUVECs. Furthermore, THZ1 dose-dependently reduced VEGFR2 expression and induced apoptosis by enhancing the cleavages of PARP and caspase-3 and-7 in HUVECs. This data (Figure 3) demonstrated that THZ1 suppressed CDK7 downstream RNAPII signaling, which led to a decrease in the effect of CDK7-associated transcription activity on angiogenic VEGFR2 expression and concurrent apoptosis in HUVECs. We concluded that THZ1 blocked the active site of CDK7 without altering CDK7 expression for suppressing downstream RNAPII phosphorylation at serine 5 and 7, which diminished the survival and angiogenesis of the endothelial cells. 

THZ1-mediated CDK7 inhibition diminished the expression of VEGFR2, a most critical angiogenic receptor, and therefore, we further examined the effect of THZ1 on angiogenesis in in vitro and in vivo systems. Figure 4A presents the results of the tube formation assay and in vivo Matrigel plug assay. As illustrated in Figure 4A, THZ1 suppressed VEGF-activated angiogenic tube formation in HUVECs in vitro. Additionally, Figure 4B illustrates that THZ1 suppressed VEGF-stimulated angiogenesis in the Matrigel plug assay in vivo. These results revealed that CDK7 inhibition blocked endothelial cell angiogenesis both in vitro and in vivo. 

To further investigate the impact of CDK7 on endothelial cell angiogenesis, CDK7 knockdown by siRNA (siCDK7) was performed. Figure 4C reveals the knockdown efficacy of siCDK7 in HUVECs, as confirmed through Western blotting. Moreover, siCDK7, but not scrambled siRNA, decreased the endothelial cell angiogenic activity in the tube formation assay (Figure 4D). Our findings indicate that the results of CDK7 knockdown were consistent with THZ1-mediated inhibition of angiogenic activity, indicating that CDK7-related downstream transcriptional activity was critical for regulating the angiogenic activity of the endothelium. 

### 3.4. THZ1 Dose-Dependently Inhibits VEGF Secretion and Expression in Caki-2 and 786-O Cells 

Approximately 50% of patients with clear cell RCC carried *VHL* gene mutations and had deregulated hypoxia-inducible factor-related downstream genes, including *VEGF* [21]. Antiangiogenesis therapy is the mainstay of metastatic RCC treatment. THZ1 has antitumor effects on several human cancers [22,26,27].

On the basis of the linkage of CDK7 and angiogenic activity of endothelial cells in this study, we next evaluated the antiangiogenic effect on RCC cells (Figure 5). Our results indicated that THZ1 dose-dependently suppressed VEGF secretion and expression in Caki-2 (Figure 5A) and 786-O cells (Figure 5B) after 24 and 48 h of treatment, indicating the potential application of THZ1 for antiangiogenic therapy of RCC.

### 3.5. THZ1 Constrains Tumor Growth of Human RCC Cells (Caki-2 and 786-O) with the Concurrent Suppression of Angiogenesis in a Xenograft Mouse Model

CDK7 inhibition significantly inhibited the angiogenic activity of the endothelium and VEGF secretion by RCC cells. Therefore, the effect of THZ1 on RCC tumor growth and angiogenesis in vivo was evaluated A xenograft mouse model of human RCC tumor (Caki-2 and 786-O) was used for the study. The mice were injected with saline or THZ1 intraperitoneal for four weeks. The appearance of tumors and tumor volume over time are presented in Figure 6A. THZ1 significantly inhibited the growth of RCC tumors. Moreover, the effect of THZ1-induced CDK7 inhibition on RCC angiogenesis was further evaluated by determining the expression of CD31, an endothelial marker, through immunohistochemical staining (Figure 6B) and Western blot analysis (Figure 6C). Immunostaining revealed that THZ1 decreased CD31 expression in the stroma between tumor cell clusters (Figure 6B), suggesting induction of angiogenesis. Additionally, Western blot analysis revealed reduced CD31 expression levels in Caki-2 and 786-O tumors exposed to THZ1 (Figure 6C). These data demonstrated that THZ1 can diminish angiogenesis, restraining the growth of RCC xenograft tumors. However, the underlying mechanism of the antiangiogenic effect on human RCC in vivo through THZ1-induced CDK7 inhibition requires further exploration. 

## 4. Discussion

CDK7 plays a dual role, namely in transcription and cell cycle regulation. It is the kinase subunit of the general transcription factor (TFIIH) that phosphorylates the CTD of RNAPII and is essentially required for transcription in cell physiology. It phosphorylates RNAPII CTD at serine 5 and 7 to participate in the initiation and elongation of transcription [8]. However, the association between CDK7 and angiogenesis remains unclear. Our results demonstrated that VEGF, a primary angiogenic factor, dose-dependently enhanced the expression of CDK7 and the transcriptional activity of RNAPII by increasing the expression and phosphorylation of RNAPII at serine 5 and 7. We found that VEGF-mediated angiogenesis was associated with the upregulation of CDK7 expression, and thus with an increase in CDK7 kinase activity and induction of RNAPII serine 5 phosphorylation at 5 and 7, which in turn activates CDK7-stimulated transcriptional activity. Furthermore, THZ1, a selective covalent CDK7 inhibitor, specifically suppressed the VEGF-activated proliferation and migration of endothelial cells. Additionally, THZ1 induced apoptosis in endothelial cells with a concurrent suppression of VEGFR2 expression, the critical angiogenic regulator. Moreover, THZ1 suppressed the angiogenic activity of endothelial cells in the in vitro capillary tube formation assay and in vivo Matrigel plug assay. To confirm the linkage between CDK7 and angiogenesis, we confirmed that CDK7 knockdown diminished angiogenesis. 

Our study shows that VEGF stimulation time-dependently enhances the expression of CDK7 on upregulating the transcriptional activity of RNAPII in endothelium. However, the molecular mechanism regarding how VEGF activates CDK7 activity and expression for increasing RNAPII-mediated transcriptional activity in endothelium during angiogenesis remains unknown. Promoter-proximal pausing is a crucial step in the RNAPII transcription cycle for thousands of human genes and organizes a rate-limiting step in mRNA synthesis [8,30]. Previous studies provided evidence that VEGF stimulates RNAPII pause release for amplifying transcription by stimulating acetylation of ETS1, a master endothelial cell transcriptional regulator, to allow angiogenesis [31]. In addition, CDK7 is an important component in blocking RNAPII promoter-proximal pausing for gene transcription [7,24,26,27]. Therefore, VEGF may also activate and enhance the expression of CDK7 on facilitating the promoter-proximal pausing release of RNAPII to enable angiogenesis in the endothelium. Furthermore, the molecular mechanism of VEGF-activated upregulation of CDK7 expression is unknown. We guessed that VEGF and CDK7 may reciprocally increase their expression of each other for amplifying the angiogenic response during tumor progression. However, the transcription of angiogenesis-related genes regulated by VEGF-CDK7- RNAPII signaling axis remains to be further elucidated.

In this study, we investigated that CDK7 inhibition by THZ1 significantly inhibited VEGF-stimulated angiogenic activity of endothelium in vitro and VEGF-activated angiogenesis in Matrigel-plug in vivo. Whereas, there are so many growth factors involved in regulation of angiogenesis [32,33], we cannot make sure that the suppressing effect of THZ1 is specific or just selectively on VEGF rather than other growth factors. Therefore, the action of other angiogenic growth factors or inhibitors on modulation of CDK7 expression and transcriptional activity is worth further study for clarifying the specific effect of THZ1 on angiogenesis. Furthermore, because CDK7 is a dual regulator in transcription and cell cycle regulations, which activities are important in angiogenesis and whether and how THZ1 affects CDK7-mediated transcription and/or cell cycle control during angiogenesis is unknown. Moreover, there are at least 20 CDKs that have been found in mammalian cells [34], but the correlation and discussion of CDKs in angiogenesis are lacking. A valuable study showed that CDK5, which is uniquely involved in neuronal development rather than in cell cycle control, participates in the regulation of the migration of endothelium during angiogenesis. CDK5-stimulated Rac1 activity contributes to control the organization of actin cytoskeleton for the formation of lamellipodia but without affecting the function of focal adhesions or microtubules during endothelial migration [20]. Furthermore, during angiogenesis, pleiotrophin can motivate endothelial migration via binding with receptor protein tyrosine phosphatase beta/zeta and ανβ3 integrin. CDK5 activation in the modulation of angiogenic migration of endothelium is mediated by pleiotrophin-receptor protein tyrosine phosphatase beta/zeta signaling [35]. Otherwise, CDK1, a well-known cell cycle and apoptosis regulator, was reported to be overexpressed in pathological retinal angiogenesis. The silence of CDK1 suppressed HUVEC’s proliferation, migration, and tube formation, which is dependent on P21- and P53-mediated G2/M phase cell cycle arrest and apoptosis induction [36]. However, whether CDK7 is action like CDK1 and CDK5 on modulation of angiogenesis that is worthy of further investigation. 

Although targeting CDK7 effectively inhibited the growth of human malignancies [7,22,24,25,27], THZ1 has antitumor effects on several human cancers [22,26,27]. In addition to the antiangiogenic effect of THZ1, we further investigated the antitumor effect of THZ1 on human clear cell RCC, which was reported to be closely associated with angiogenesis. We demonstrated that THZ1 suppressed the expression and production of VEGF in RCC cells, and that THZ1 has antitumor effects on human RCC xenografts with the concurrent suppression of the angiogenic marker CD31. 

Targeting the VEGF pathway has been first-line therapy in metastatic RCC. Current targets of antiangiogenic therapy targeting VEGF or VEGFR lead to substantial side effects due to cross-reactivity with physiological both physiological and pathological angiogenesis. THZ1 has been report to elicit antitumor efficacy with good tolerability in pre-clinical studies. [22,26,27]. The suppression of angiogenesis of THZ1 served as a new mechanism of its antitumor activity. This is the first study that evaluated the role and function of CDK7 in angiogenesis and human RCC. Our results indicated that targeting CDK7 is a promising strategy for treating metastatic RCC. The underlying mechanism involved in CDK7-activated transcription and RCC tumor growth should be further studied.

There are some limitations in our study. First, THZ1, as a CDK7 covalent inhibitor, also inhibits the closely related kinases CDK12 and CDK13 (CDK12/13) [37]. The exact mechanisms for the anti-angiogenesis of THZ1 should be further clarified in future studies. Second, despite controlling dysregulated tumor-related angiogenesis as a promising target to inhibit cancer progression [38], tumor microenvironment comprises numerous signaling molecules and pathways that influence the angiogenic response. Our study validates CDK7 as a novel antiangiogenic target and a promising strategy for treating metastatic RCC. Nevertheless, the underlying mechanism involved in CDK7-induced antiangiogenesis and tumor inhibition of RCC should be further studied.

## 5. Conclusions

Our work elucidated the reciprocal activation of VEGF and CDK7 in angiogenesis, because VEGF promoted CDK7 expression and CDK 7 inhibition suppressed angiogenic activity of endothelium and VEGF secretion of RCC cells, indicating CDK7 as a pivotal new regulator of angiogenesis in both endothelial cells and RCC cells, and suggesting that CDK7 pharmacologically is a novel target for antiangiogenic therapy and provides the basis for a new therapeutic application of the CDK7 inhibitor, THZ1, as an antiangiogenic agent.

## Figures and Tables

**Figure 1 cells-08-01469-f001:**
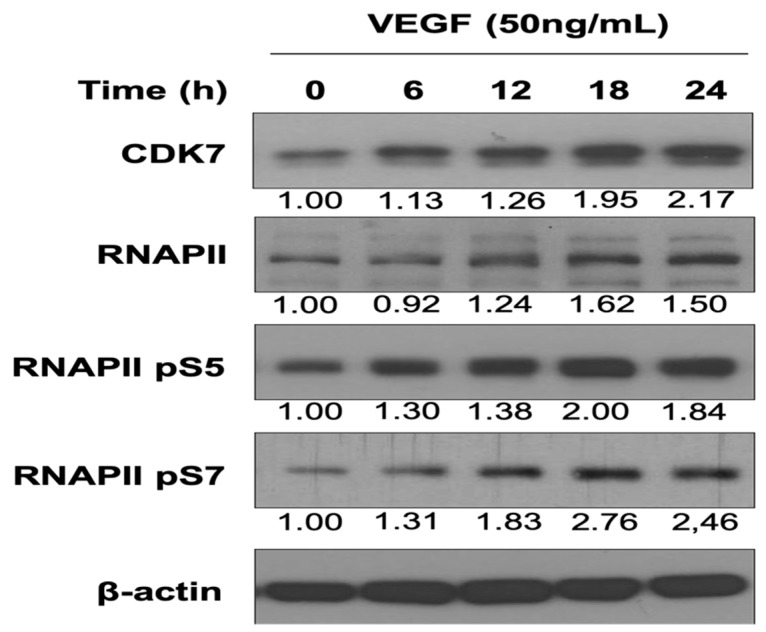
Vascular endothelial growth factor (VEGF) increased Cyclin-dependent kinase 7 (CDK7) and RNA polymerase II (RNAPII) expression and downstream RNAPII phosphorylation at serine 5 and 7 in human umbilical vein endothelial cells (HUVECs). HUVECs were treated with 50 ng/mL VEGF for 6–24 h. The cells were then harvested and subjected to Western blotting analysis using specific CDK7, RNAPII, RNAPII pS5, and RNAPII pS7 antibodies. Numbers under the plots indicate the fold change of protein level relative to their untreated control. Similar results were obtained in at least three independent experiments.

**Figure 2 cells-08-01469-f002:**
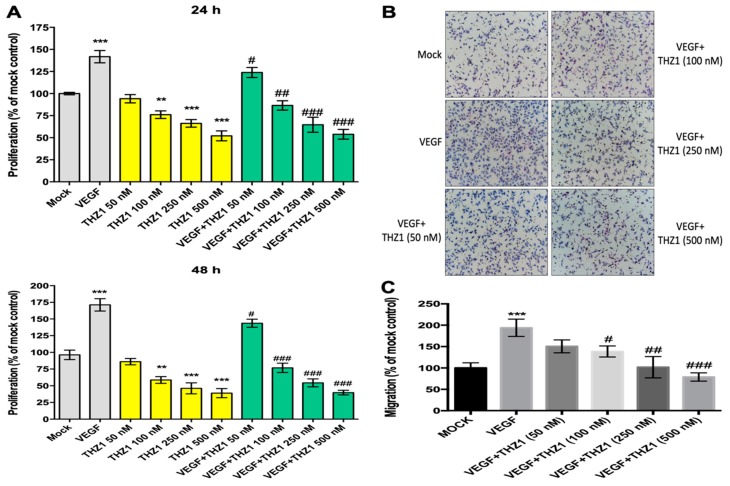
THZ1 suppressed the proliferation and chemotactic migration of HUVECs in a dose-dependent manner. (**A**) HUVECs (5 × 10^3^ cells) in a 96-well-plate were treated with VEGF (50 ng/mL) and various concentrations of THZ1 (50, 100, 250, 500 nM) for 24 and 48 h. After incubation, HUVEC proliferation was analyzed by the water-soluble tetrazolium 1 (WST-1) assay. Data are presented as mean ± standard deviation. ** *p* < 0.01 and *** *p* < reach untreated mock control group; # *p* < 0.05, ## *p* < 0.01, and ### *p* < 0.001 compared to the each VEGF (50 ng/mL) treated group (**B**,**C**). The chemotactic motility of HUVECs was assayed by the Transwell assay. Cell suspension (100 µL) containing 2 × 10^4^ cells of HUVECs was loaded into each upper well. VEGF (10 ng/mL) with or without various concentrations of THZ1 (50, 100, 250, and 500 nM) in 5% fetal bovine serum (FBS)-containing M199 were placed in the lower wells. The chamber was incubated at 37 °C for 4 h. After incubation, the cells were fixed and stained. Chemotaxis was observed by optical microscopy at 100× magnification and quantified to detect the cells that migrated to the lower side of the filter. *** *p* < 0.001 compared to the untreated mock control group; # *p* < 0.05, ## *p* < 0.01, and ### *p* < 0.001 compared to the VEGF (10 ng/mL)-treated group. Similar results were obtained in at least three independent experiments.

**Figure 3 cells-08-01469-f003:**
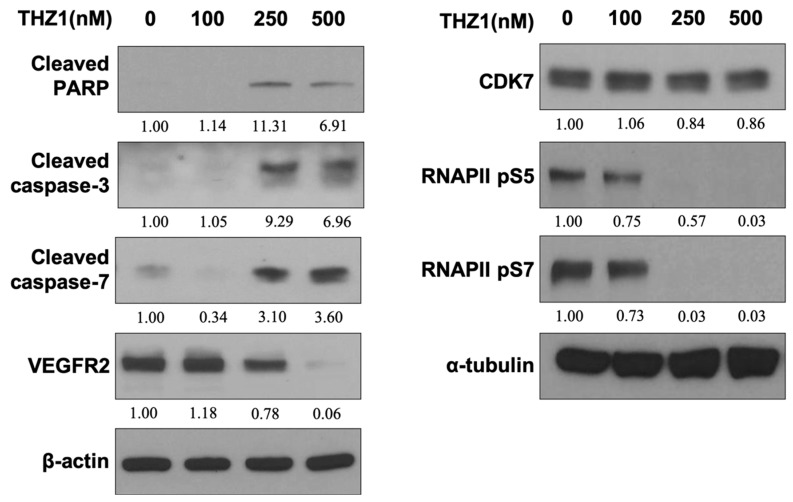
THZ1 induced apoptosis and suppressed VEGFR2 expression and RNA polymerase II (RNAPII) C-terminal domain (CTD) phosphorylation at serine 5 and 7 in HUVECs**.** HUVECs were treated with various concentrations of THZ1 (100–500 nM) for 48 h. Cell lysates were harvested and subjected to Western blotting using specific antibodies against cleaved caspase-3, cleaved caspase-7, cleaved PARP, VEGFR2, CDK7, RNAPII p55, and RNAPII pS7. Numbers under the plots indicate the fold change of protein level relative to their untreated control. The illustrated results are representative of at least three independent experiments.

**Figure 4 cells-08-01469-f004:**
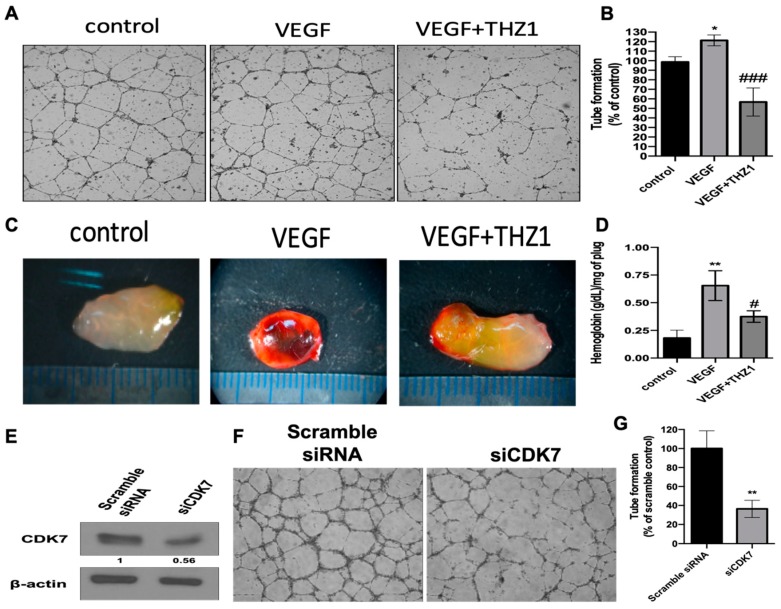
THZ1-induced CDK7 inhibition or CDK7 knockdown suppressed angiogenesis in vitro and in vivo. (**A**,**B**) THZ1 reduced VEGF-activated capillary-like tube formation in HUVECs. HUVECs were plated on Matrigel-coated wells and then treated with VEGF (10 ng/mL) in the presence or absence of THZ1 (250 nM). After 16–24 h incubation, HUVEC’s tube formation on Matrigel was photographed (**A**) and further quantified by counting tube density (**B**). * *p* < 0.05 compared to the control group (DMSO); ### *p* < 0.001 compared to the VEGF (10 ng/mL)-treated group. (**C**,**D**) THZ1 diminished VEGF-activated angiogenesis in the in vivo Matrigel plug assay. Matrigel (500 µL) containing VEGF (100 ng) or VEGF (100 ng) plus THZ1 (500 nM) was subcutaneously injected into the ventral area of six-week-old FVB mice (*n* = 3). After 5 days, Matrigel plugs were excised and photographed (**C**). Furthermore, hemoglobin content of recovered Matrigel plugs was determined (**D**). ** *p* < 0.01 compared to the control group (DMSO); # *p* < 0.05 compared to the VEGF (10 ng/mL)-treated group. (**E**–**G**) siCDK7 decreased CDK7 expression and angiogenic tube forming activity in HUVECs. HUVECs were transfected with scrambled or CDK7-specific siRNAs (siCDK7; cdk-466) for 24 h. The cells were then harvested and subjected to Western blotting analysis for checking CKD7 expression (**E**). Numbers under the plots indicate the fold change of protein level relative to their control. Further, the transfected cells were seeded onto a Matrigel for verifying and quantifying the activity of tube formation (**F**,**G**). ** *p* < 0.01 compared to the scramble siRNA control group. Similar results were obtained in two independent experiments.

**Figure 5 cells-08-01469-f005:**
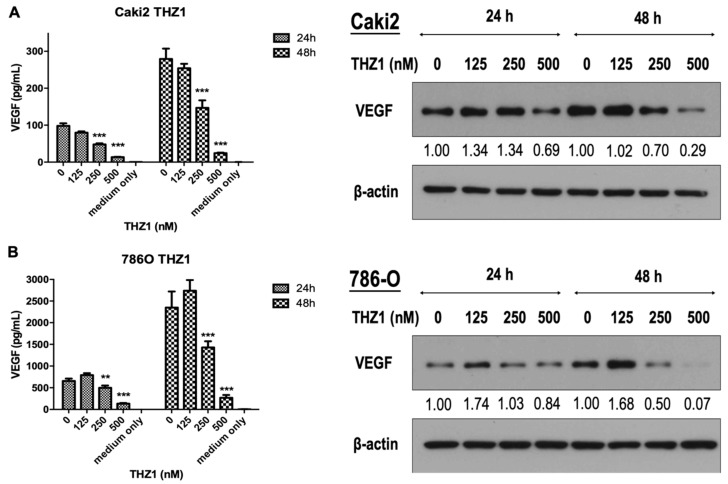
THZ1 dose-dependently inhibited VEGF secretion and expression in Caki-2 and 786-O cells. (**A**) Caki2 and (**B**) 786-O cells were treated with various concentrations of THZ1 (125, 250, and 500 nM) for 24 and 48 h. After treatment, the conditioned media were collected for determining VEGF secretion using enzyme-linked immunosorbent assay (ELISA), and the cell lysates were subjected for VEGF expression analysis by Western blotting using antibody against human VEGF. ** *p* < 0.01 and *** *p* < 0.001 compared to the control group (DMSO). Numbers under the plots indicate the fold change of protein level relative to their control. Similar results have been obtained in two independent experiments.

**Figure 6 cells-08-01469-f006:**
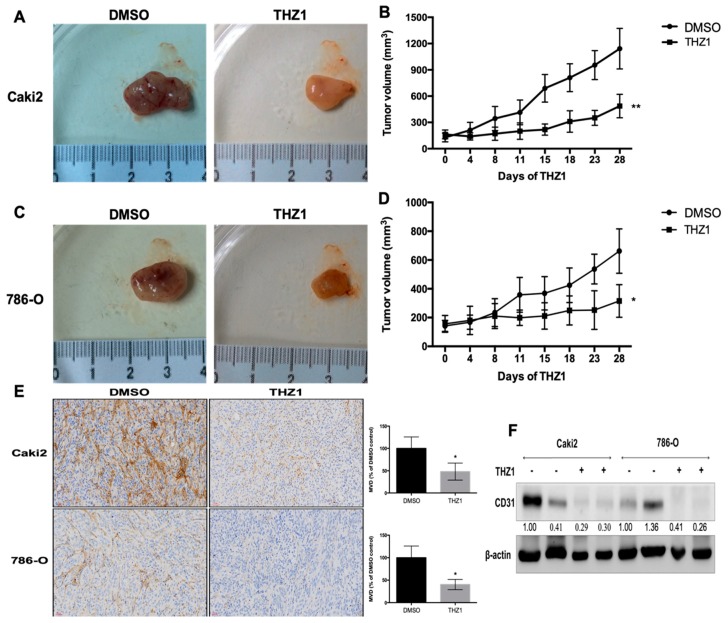
THZ1 inhibited the tumor growth of human C-terminal domain (RCC) cells (Caki-2 and 786-O) with concurrent suppression of angiogenic activity in a xenograft mouse model. Caki-2 and 786-O tumor-bearing mice were intraperitoneally injected with DMSO (*n* = 7) or THZ1 (*n* = 8) for 4 weeks. (**A**,**C**) Tumors were photographed after harvest. (**B**,**D**) Tumor volumes were measured to evaluate the antitumor response to THZ1. Tumor volumes are presented as mean ± standard deviation. * *p* < 0.05 and ** *p* < 0.01 compared to the control group (DMSO). (**E**,**F**) THZ1 suppressed CD31 expression in the human RCC tumor xenograft. Distribution and expression of CD31 was determined by immunohistochemical staining (**E**) and Western blot analyses (**F**). * *p* < 0.05 compared to the control group (DMSO).

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
