# Peer review of "Suppression of Angiogenesis by Targeting Cyclin-Dependent Kinase 7 in Human Umbilical Vein Endothelial Cells and Renal Cell Carcinoma: An In Vitro and In Vivo Study"

_cells, 2019, doi:10.3390/cells8111469_

Round 1

Reviewer 1 Report

This study by Shi et al has examined the role of CDK7 in angiogenesis and RCC.  This paper presents some nice results but there are some issues that need to be clarified and corrected before publication.

You have not used consistent doses of THZ-1 in your in vitro experiments.  Initial proliferation experiments use different doses to your Western blot analysis, what was the reason for this?

Many of your figures would be strengthened by quantification of changes that you are presenting.  For example figure 2B, your migration data - it doesn't look like there is a large decrease in migration until the highest concentration of THZ1.  Also in the figure legend for figure 2, you mention #p<0.05 but I can't see that symbol anywhere on your figure.

In your knockdown experiments, what was your percentage knockdown that you showed in your Western blots.  There is no mention of the siRNA experiments in your methods and how the transfection took place.

It would be nice to see some extra quantification in the tube forming assays shown in figures 4A and C.  From the images that you have presented in panel A, it is not convincing that VEGF is indeed showing an increase in tube formation compared to the mock panel in this figure.  You describe that the cells were place in media, you need to clarify what this media is - if it still contains ECGS and 20% serum then this may be the reason that VEGF doesn't appear to be stimulating tube formation.  Could you perhaps quantify your tube formation images to more clearly demonstrate an effect.

Your interpretation of the in vivo Matrigel plug assay is that you have a decrease in VEGF induced angiogenesis in this model.  Based on the results you have presented I think that you are assuming this based solely on the fact that your plugs are different sizes upon excision from the mouse.  This is a big stretch.  It would be nice to see what is present inside the plugs before you make this assessment, you have a CD31 antibody so it would be nice to see some staining of these plugs.

In figure 6 your panels are labelled incorrectly and the figure legend does not match what is presented.  Additionally, in your legend you have not presented n numbers just n=?.  I am unsure what the logic behind performing both immunohistochemistry for CD31 as well as Western blot analysis. It would be nice to see some quantification of this data to further support your observations.

I am a little bit concerned that your reported n are low - in some cases only 2 and in the mice only 3 per group.  In some cases this needs to be increased to allow proper quantification of the experimental data.

This study presents some interesting results but the data presented needs to be improved.

Reviewer 2 Report

The manuscript by Shi et al. describes the suppression of angiogenesis by targeting cyclin 3 dependent kinase 7 in human umbilical vein endothelial cells and renal cell carcinoma. It is a very interesting manuscript showing promising results. Please see my specific comments bellow.

1. Page 4, line 127. Statistic analysis heading. The authors could describe the statistics analysis more detailed. Sometimes, it quit confusing, i.e. “Comparisons were considered statistically different if p < 0.05. one-way analysis of variance”.

2. Methods section, page 4, lines 147-148. “The cell suspensions (40 µL each) containing various concentrations of THZ1 or VEGF”. Please, specify which concentrations.

3. When do cells has the best tubular formation? In the methods, the authors described 16-24h. However, from what time does tubular formation occur?

4. Although the results are well presented, the discussion section was poorly explored. Since different results were showed, authors should provide a more detailed discussion about the results.

5. Please, provide the study limitation in discussion section.

6. How about the conclusion? The author should provide a conclusion based on their results.
